KIT is involved in melanocyte proliferation, apoptosis and melanogenesis in the Rex Rabbit

Hu Shuaishuai 1
Chen Yang 1 2
Zhao Bohao 1
Yang Naisu 1
Chen Shi 1
Shen Jinyu 1
Bao Guolian 3
Wu Xinsheng xswu@yzu.edu.cn 1 2
1 College of Animal Science and Technology, Yangzhou University , Yangzhou , Jiangsu , China
2 Joint International Research Laboratory of Agriculture & Agri-Product Safety, Yangzhou University , Yangzhou , Jiangsu , China
3 Animal Husbandry and Veterinary Research Institute, Zhejiang Academy of Agricultural Sciences , Hangzhou , Zhejiang , China
Silva Pedro
Electronic publication date: 2020 Jun 18
Publication date: 2020
Volume: 8
Electronic Location ID: e9402
Received 2019 Nov 27; Accepted 2020 Jun 1
Copyright: ©2020 Hu et al.
Copyright year: 2020
Copyright holder: Hu et al.
License: This is an open access article distributed under the terms of the Creative Commons Attribution License, which permits unrestricted use, distribution, reproduction and adaptation in any medium and for any purpose provided that it is properly attributed. For attribution, the original author(s), title, publication source (PeerJ) and either DOI or URL of the article must be cited.
License URL: https://creativecommons.org/licenses/by/4.0/

Keywords: KIT, Melanocyte, Melanogenesis, Proliferation, Apoptosis

Funding: National Natural Science Foundation of China 31702081 Modern Agricultural Industrial System Special Funding CARS-43-A-1 Science and Technology Major Project of New Variety Breeding (Livestock and Poultry) of Zhejiang Province, China 2016C02054-10 The study was supported by the National Natural Science Foundation of China (Grant No. 31702081), the Modern Agricultural Industrial System Special Funding (CARS-43-A-1), and the Science and Technology Major Project of New Variety Breeding (Livestock and Poultry) of Zhejiang Province, China (2016C02054-10). The funders had no role in study design, data collection and analysis, decision to publish, or preparation of the manuscript.

==============================
Background

Melanocytes play an extremely important role in the process of skin and coat colors in mammals which is regulated by melanin-related genes. Previous studies have demonstrated that KIT is implicated in the process of determining the color of the coat in Rex rabbits. However, the effect of KIT on the proliferation and apoptosis of melanocytes and melanogenesis has not been clarified.

Methods

The mRNA and protein expression levels of KIT were quantified in different coat colored rabbits by qRT-PCR and a Wes assay. To identify whether KIT functions by regulating of melanogenesis, KIT overexpression and knockdown was conducted in melanocytes, and KIT mRNA expression and melanin-related genes TYR, MITF, PMEL and DCT were quantified by qRT-PCR. To further confirm whether KIT influences melanogenesis in melanocytes, melanin content was quantified using NaOH lysis after overexpression and knockdown of KIT. Melanocyte proliferation was estimated using a CCK-8 assay at 0, 24, 48 and 72 h after transfection, and the rate of apoptosis of melanocytes was measured by fluorescence-activated cell sorting.

Results

KITmRNA and protein expression levels were significantly different in the skin of Rex rabbits with different color coats (P < 0.05), the greatest levels observed in those with black skin. The mRNA expression levels of KIT significantly affected the mRNA expression of the pigmentation-related genes TYR, MITF, PMEL and DCT (P < 0.01). Melanin content was evidently regulated by the change in expression patterns of KIT (P < 0.01). In addition, KIT clearly promoted melanocyte proliferation, but inhibited apoptosis.

Conclusions

Our results reveal that KIT is a critical gene in the regulation of melanogenesis, controlling proliferation and apoptosis in melanocytes, providing additional evidence for the mechanism of pigmentation of animal fur.

Introduction

Rex rabbits have an important and archetypal fur, which is represented in China by a considerable number of colors, including black, white, chinchilla, brown, gray, gray-yellow, etc. (Fig. 1). Therefore, it serves as an important experimental animal model of pigmentation, which is instructive in defining mammalian coat colors. The diverse range of hair and skin colors in mammals can be attributed to the type and quantity of pigment. However, the formation of pigment is regulated by the interaction of different genes and environmental factors (Sturm, Teasdale & Box, 2001). Melanin produced by melanocytes is one of a number of pigments that are regulated by melanin-related genes, which determine the production of phenotypes of diverse hair and skin color in mammals. Currently, there are 128 genes of pigmentation associated with phenotypes in humans (Baxter et al., 2019). More than 170 pigmentation-related genes associated with coat color phenotype have been identified in mice (Bennett & Lamoreux, 2003). Moreover, at least 31 genes that have a significant effect on coat color have been identified in mink (Song et al., 2017). Therefore, it might be reasonable to conclude that the coat color of Rex rabbits is regulated by various pigmentation-related genes.

Among melanin-related genes, KIT is pivotal in the melanogenesis signaling pathway, and mutations or deletion of KIT can cause different hair and skin colors in mammals. KIT is a proto-oncogene, classified as a member of the tyrosine kinase receptor family, the product of which is the type III tyrosine kinase. In addition, KIT plays an important role in the melanogenesis pathway. It was reported that melanogenesis can be enhanced by stem cell factor/c-kit signaling in normal human epidermal melanocytes exposed to norepinephrine (Lan et al., 2008). In previous studies, the occurrence of melanomas was found to be often associated with a mutation in KIT (Curtin et al., 2006; Sakaizawa et al., 2015). It has been reported that a single A →G base missense mutation in exon 13 causes differential expression of KIT in Liaoning Cashmere goats resulting in different coat colors (Li et al., 2017). The duplication of chromosome 6 and aberrant insertion on chromosome 29 leads to coat color variations in white Galloway cattle and white Park cattle (Brenig, Beck & Floren, 2013). White spotting is caused by a frameshift mutation of KIT in the Arabian camel (Holl et al., 2017). Furthermore, melanogenesis is additionally regulated by a number of key genes, which control melanocytes to further generate melanin, such as the key melanogenic enzyme genes TYR and DCT (Hearing, 2011; Yasumoto et al., 1997), and critical melanosomal protein PMEL (Chen et al., 2018; Ji et al., 2018). In addition, KIT has a critical influence on melanin deposition. It is known that KIT regulates cell migration, survival, proliferation and differentiation in melanocytes (Garrido & Bastian, 2010; Grichnik, 2006) and interacts with MITF, a crucial gene that is implicated in the formation of melanin and further regulates the development of melanocytes (Mizutani, Hayashi & Imokawa, 2010; Reiko et al., 2004; Wen et al., 2010). Our previous studies have indicated that KIT is involved in the formation of different coat colors in the Rex rabbit. However, the role of KIT in melanogenesis, proliferation and apoptosis in melanocytes remains largely unexplored. Furthermore, it is necessary to determine whether the expression of KIT can indirectly affect melanogenesis-related genes.

Figure 1 The Rex rabbits of different coat colors.

(A) White, (B) black, (C) chinchilla, (D) brown, (E) gray, (F) gray-yellow.

In the present study, the mRNA expression levels of KIT and melanin-related genes in diverse coat colors of the Rex rabbit were measured by quantitative real-time PCR (qRT-PCR). In order to reveal the role of KIT in melanocyte proliferation, apoptosis and melanogenesis, the change in rabbit melanocytes was detected by overexpression and knockdown of KIT. The results of the study provide an important theoretical basis for the additional study of mammal skin and hair color.

Materials and methods

Ethics statement

This study was conducted in accordance with the recommendations of the Animal Care and Use Committee at Yangzhou University (Jiangsu, China), who approved the experimental procedures (Yangzhou, China, 24 November 2018, No. 201810025) and were strictly implemented according to the regulations for experimental animals.

Experimental animals and sample collection

Animals were maintained in an ordinary housing facility in keeping with the national standard Laboratory Animal Requirements of Environment and Housing Facilities (GB 14925-2001). The care of laboratory animals and the experimental animal surgery conformed to the Jiangsu Administration Rule of Laboratory Animals. The rabbits were anesthetized using an intraperitoneal injection of sodium pentobarbital. Rex rabbits (black, white, chinchilla, brown, gray and gray-yellow) were provided by Zhejiang Yuyao Xinnong Rabbit Industry Co., Ltd. Four rabbits (male and female in equal numbers) of each coat color were raised within the same environment. Ninety days after birth, dorsal skin samples of identical size (one cm2) and anatomical location were harvested from each rabbit for RNA and protein extraction. The wounds were treated topically with iodophor after sample collection. All rabbits made a full recovery from surgery and were free to move after two hours.

Cloning of the KIT gene

Primers for three specific 5′ untranslated regions (UTRs), coding sequence (CDS), and two 3′ UTRs of KIT were synthesized (Table 1). A rapid amplification of cDNA ends (RACE) assay was conducted in accordance with the kit instructions (Invitrogen & Clontech, Carlsbad, CA, USA). The coding sequence of the KIT gene was acquired based on the results of 5′ and 3′ RACE assays. KIT cDNA was cloned and reconstructed into pcDNA3.1 or pcDNA3.1-Myc vectors with QuickCut Restriction Enzyme Hind III and EcoR I (Takara, Beijing, China) using a ClonExpress II One Step cloning kit (Vazyme, Nanjing, China), respectively.

Table 1 Primers used in the KIT RACE assay.

Primers for three specific 5′untranslated regions (UTRs), coding sequence (CDS), and two 3′UTRs of KIT were synthesized.

Name	Sequence (5′ to 3′)	Experiment	
B065-1 (GSP1)	GGAGTTTCCCAGGAGTCGGC	5′ RACE	
B065-2 (GSP2)	GCTCGGTTTCAGCATCTTCACA	5′ RACE	
B065-3 (GSP3)	GCCTTGGAACTGGGACTGAG	5′ RACE	
C047-1	CCGACGGCTGCTTGTTTTAG	3′ RACE	
C047-2	CTCTTCTTGTTGCTGTGGG	3′ RACE	
KIT-R	ATGCTTTTTTTCCGCCAAAGA	CDS	
KIT-F	GTCTTCATGCACGAGCAGGG	

Rabbit melanocyte culture and overexpression transfection of KIT

Rabbit melanocytes were isolated from the dorsal skin tissue of Rex rabbits as described by Chen et al. (2019). The cells were cultured in M254 medium (Gibco, Carlsbad, CA, USA) supplemented with 1% human melanocyte growth supplement-2 (HMGS-2, Gibco, Carlsbad, CA, USA) and maintained in an incubator at 37 °C in an atmosphere containing 5% CO2. The seeded cells were cultured in 6-well plates to 70–90% confluence, then overexpression of KIT achieved using Lipofectamine™ 3000 Reagent (Invitrogen, Carlsbad, CA, USA), in accordance with the manufacturer’s instructions. A 2 µg quantity of KIT DNA plasmid in Opti-MEM™ medium (Gibco, Carlsbad, CA, USA) was added to 6 µL Lipofectamine™ 3000 diluted with Opti-MEM™ then incubated for 10–15 mins at room temperature. The DNA-lipid complex was added cells and incubated for 48 h with M254 medium, prior to qRT-PCR analysis.

Knockdown of KIT using siRNA

siRNAs of KIT and the negative control were purchased from Shanghai GenePharma Co., Ltd (Table 2). The cells were grown to 70–90% confluence, then knockdown of KIT was conducted using Lipofectamine™ 3000 (Invitrogen, Carlsbad, CA, USA), in accordance with the manufacturer’s instructions. 0.264 µg (1 µL) siRNA-KIT and 3 µL Lipofectamine 3000 were used in each well of 6-well plates. The transfection method was the same as that used for overexpression transfection of KIT. Transfected cells were analyzed using qRT-PCR after 36 h.

Table 2 Primer sequences of siRNA-KIT.

siRNAs of KIT and the negative control were purchased from Shanghai GenePharma Co., Ltd.

Name	Sequence (5′ to 3′)	
Negative Control	Forward: UUCUCCGAACGUGUCACGUTT	
Reverse: ACGUGACACGUUCGGAGAATT	
siRNA-158	Forward: GGUUCUCGCUGGAGUGCAUTT	
Reverse: AUGCACUCCAGCGAGAACCTT	
siRNA-791	Forward: GCUGGCAUCAGGGCGACUUTT	
Reverse: AAGUCGCCCUGAUGCCAGCTT	
siRNA-921	Forward: CCUUGAAAGUCGUAGAUAATT	
Reverse: UUAUCUACGACUUUCAAGGTT	

RNA isolation and quantitative real-time PCR (qRT-PCR)

Total RNA was isolated from skin tissues and transfected cells using an RNAsimple Total RNA kit (Tiangen Biotech (Beijing) Co., Ltd.). The quality and concentrations of RNA samples were quantified from measurements of optical density (OD) (OD = 260/280) using a NanoDrop 1000 (ThermoFisher Scientific, New York, USA). One µg total RNA was used to synthesize cDNA using a Super RT cDNA kit (Tiangen Biotech Co., Ltd.). Quantitative real-time PCR was conducted using ChamQ™ SYBR® qPCR Master Mix (Vazyme, Nanjing, China), with 1 µL cDNA in each well and 1 µL primer with a 0.5 µmol primer final concentration. Data were analyzed using QuantStudio® 5 software (Applied Biosystems; Thermo Fisher Scientific, Foster City, CA). Each sample was measured four times and relative expression of target genes computed using the 2− ΔΔCt method (Schmittgen & Livak, 2008) after normalization to GAPDH as the endogenous control. KIT, MITF, TYR, DCT, PMEL and GAPDH gene sequences were obtained from NCBI for design of the primers (Table 3).

Table 3 Primer sequences for quantitative real-time PCR.

KIT, MITF, TYR, DCT, PMEL and GAPDH gene sequences were obtained from NCBI for design of the primers.

Genes	Sequence (5′ to 3′)	Product length (bp)	
KIT	Forward: GGAGTTTCCCAGGAGTCGGC	139	
Reverse: GCTCGGTTTCAGCATCTTCACA	
MITF	Forward: GCCTTGGAACTGGGACTGAG	142	
Reverse: CCGACGGCTGCTTGTTTTAG	
TYR	Forward: CTCTTCTTGTTGCTGTGGG	156	
Reverse: GCTGAGTAGGTTAGGGTTTTC	
DCT	Forward: ATTCTGCTGCCAATGACCC	154	
Reverse: AACGGCACCATGTTATACCTG	
PMEL	Forward: GTCAGCACCCAGCTTGTCA	130	
Reverse: GCTTCATTAGTCTGCGCCTGT	
GAPDH	Forward: CACCAGGGCTGCTTTTAACTCT	141	
Reverse: CTTCCCGTTCTCAGCCTTGACC	

Isolation of proteins and Wes

Skin and cell lysates were obtained using RIPA lysis buffer (PPLYGEN, Beijing, China). Protein concentrations were quantified using an enhanced BCA protein assay kit (Beyotime, Shanghai, China). A 7.5 μg protein sample in each well was analyzed using a Wes automated Western blotting system purchased from Protein Simple (Harris, 2015). Analysis was conducted using an anti-GAPDH mouse monoclonal antibody diluted 1:100 (Abcam, Cambridge, UK) and anti-KIT mouse monoclonal antibody (BBI, Beijing, China) diluted 1:100.

Melanin content measurement

Rabbit melanocytes were obtained 72 h after transfection then washed in phosphate buffered saline (PBS) (HyClone, Logan, USA) 3 times, then centrifuged at 1000 rpm for 10 min at 4 °C, the supernatant discarded after each wash. The cells were then lysed in 1 mL of 1 mol/L NaOH and incubated at 80 °C for 1 h. Finally, the melanin concentration was determined from measurements of optical density (OD) at 475 nm using an Infinite M200 Pro (Tecan, Männedorf, Switzerland) spectrophotometer.

Cell proliferation assay

Cell proliferation was evaluated using a Cell Counting Kit-8 assay (Vazyme, Nanjing, China), in accordance with the manufacturer’s instructions, as described previously by Zhao et al. (2019). Cells were harvested and seeded in 96-well plates after 16 h transfection, then the OD at 450 nm of each well was measured at 0, 24, 48 and 72 h using an Infinite M200 Pro spectrophotometer (Tecan, Männedorf, Switzerland).

Apoptosis assay

Apoptosis was measured using an Annexin V-FITC apoptosis detection kit (Vazyme, Nanjing, China). The cells were collected 48 h after transfection and sorted by fluorescence-activated cell sorting using a FACSAria SORP flow cytometer (Becton Dickinson, San Jose, CA). Each sample was measured in triplicate. Positive FITC staining fluoresces green and PI fluoresces red. Thus, living cells have only very low background fluorescence. Early apoptotic cells have strong green fluorescence only while late apoptotic cells exhibit dual red and green fluorescence.

Statistical analysis

Each experiment was performed in triplicate and analyzed using IBM SPSS v25 (SPSS Inc., Chicago, IL). Significant differences in relative gene expression were analyzed by one-way ANOVA. All values represent means ± standard deviation (SD).

Results

Expression of KIT in the skin of Rex rabbits with different colors of coat

The 5′ UTR and 3′ UTR sequences of the KIT gene were acquired using a RACE assay. The coding sequence of KIT was spliced according to the 5′ and 3′ RACE results, and the KIT cDNA sequence successfully cloned and reconstructed into a pcDNA3.1 or pcDNA3.1-Myc vector (Figs. 2A–2C). The rabbit KIT gene included a 48 bp 5′ UTR, 1,399 bp 3′ UTR and a 2,910 bp coding sequence (CDS) which was submitted to NCBI (GenBank: KY971605), and the coding sequence (CDS)of rabbit KIT RACE results was less 12 bp (4 amino acids) than the reference sequence. However, other amino acids are same, and the conserved domains are also same. The phylogenetic relationship of the assembled cDNA of KIT in Oryctolagus cuniculus to other species (Urocitellus parryii, Marmota, Jaculus jaculus, Microtus ochrogaster, Carlito syrichta and Ochotona princeps) was explored using MEGA6. The results indicated that the other six species were divided into two clades, indicating that Oryctolagus cuniculus had a more remote relationship to all other species tested than they did among themselves (Fig. 2D).

Figure 2 Cloning of the KIT gene and mRNA expression in Rex rabbits with different coat colors.

(A–C) 5′ UTR , 3′ UTR and cDNA of KIT was acquired by RACE and cloning techniques. (D) Neighbor-joining phylogenetic tree of Oryctolagus cuniculus and other species based on KIT cDNA. (E) mRNA expression of KIT gene in rabbits of different coat colors by qPCR. (F) KIT protein expression in rabbits of different coat colors was detected by Western blotting. Small letters indicate significant differences among groups (P < 0.05). WH: White, BL: Black, CH: Chinchilla, BR: Brown, GR: Gray, GY: Gray-yellow.

Subsequently, the expression levels of KIT were quantified in the different coat colors. The mRNA expression of the KIT gene was highest in rabbits with black skin, followed by chinchilla, brown, gray, gray-yellow, the lowest level observed in white skin (P < 0.05) (Fig. 2E). It was found that the protein expression levels of KIT with black skin were also higher than in rabbits with other skin colors, the lowest being white skin. KIT protein expression was in accordance with that of mRNA expression (Fig. 2F).

Melanogenesis-related genes were regulated by overexpression and knockdown of KIT

The homology of KIT in Oryctolagus cuniculus was compared with other mammals using an NCBI conserved domain search. Approximately 80% homology was found overall, the rabbit KIT protein possessing a common domain (PKC-like superfamily) that was the same as other mammals (Fig. 3A). It is known that PKC-β can regulate the activity of tyrosinase, combine with melanin in the cell membrane and further promote melanogenesis (Mochly-Rosen, 1995). To identify whether KIT can regulate melanogenesis, overexpression and knockdown of KIT was performed in melanocytes. The results indicate that knockdown of KIT using three different siRNAs resulted in expression clearly lower than that of the negative control (NC) group (P < 0.01), with siRNA-791 having the greatest effect (Fig. 3B). KIT mRNA and protein expression levels increased significantly after overexpression of KIT (P < 0.01), and decreased after knockdown of KIT (P < 0.01) (Fig. 3D). Furthermore, KIT mRNA expression and that of the melanin-related genes TYR, MITF, PMEL and DCT increased considerably when KIT was overexpressed in melanocytes (Fig. 3F), and decreased substantially after knockdown of KIT (P < 0.01) (Fig. 3E).

Figure 3 mRNA expression of melanogenesis-related genes was regulated by the change of KIT expression.

(A) The conserved domain of KIT was predicted by NCBI. (B) KIT mRNA expression levels was detected by qPCR after transfection with siRNAs-KIT. (C) mRNA expression of melanogenesis-related genes was analyzed by qPCR after knockdown of KIT. (D) mRNA expression levels of KIT was detected by qPCR when KIT was overexpressed. (E) mRNA expression levels of melanogenesis-related genes as detected by qPCR when KIT was overexpressed. ** P < 0.01, * P < 0.05.

KIT promoted melanogenesis and affected proliferation and apoptosis of melanocytes

To further confirm whether KIT affected melanogenesis in melanocytes, the level of melanin was quantified after both overexpression and knockdown of KIT. Melanin levels increased significantly compared with the control group after overexpression of KIT (P < 0.05). Conversely melanin levels clearly decreased (Figs. 4A, 4B). The proliferation of melanocytes was estimated using a CCK-8 assay 0, 24, 48 and 72 h after transfection. The results indicate that overexpression of KIT greatly promoted melanocyte proliferation from 24 to 72 h (P < 0.01), while proliferation was significantly inhibited after knockdown of KIT (P < 0.01) (Figs. 4C, 4D). Furthermore, the degree of melanocyte apoptosis was detected by fluorescence-activated cell sorting. It was found that melanocyte apoptosis was inhibited by KIT overexpression (Figs. 4E, 4F), but knockdown of KIT promoted apoptosis (Figs. 4G, 4H), indicating that KIT had a negative effect on apoptosis.

Figure 4 Melanogenesis, melanocyte proliferation and apoptosis were controlled by the change of KIT expression.

(A–B) 48 h after transfection with overexpression or knockdown of KIT, melanin content was analyzed by NaOH lysis. (C–D) Melanocyte proliferation was estimated by CCK-8 assay 24, 48, and 72 h after overexpression or knockdown of KIT. (E–F) Cell apoptosis rate in melanocyte was assessed after overexpression or knockdown of KIT. ** P < 0.01, * P < 0.05.

Melanin content and the expression of KIT have a positive correlation

To determine whether melanin content has correlation with the expression of KIT, correlation analysis was performed by Pearson Correlation in SPSS v. 22.0 software. The results clearly found a significantly positive correlation between the melanin content and the mRNA expression of KIT (P < 0.01) (Table 4). It was demonstrated that the expression of KIT has an important effect on melanin content.

Table 4 Correlation between gene expression and melanin content.

The correlation analysis between mRNA expression of KIT and melanin content was performed.

Gene expression	KIT overexpression	KIT knockdown	
Melanin content (µg/µL)	1.000**	1.000**	
Notes.

* Significant difference is denoted by *P < 0.05.

** Extremely significant difference is denoted by **P < 0.01.

Discussion

Melanin produced by melanocytes controls mammalian coat color. Differing coat colors represent a high research value and are an economically valuable trait in mammals. Different coat pigmentation is controlled by the ratio of pheomelanin to eumelanin (Greg & George, 2007; Nicola et al., 2013), the former principally involved in the production of red and yellow and the latter participating in the formation of black pigment (Ito et al., 2010; Ito, 2010; Shosuke, Kazumasa & Photobiology, 2010). It has been demonstrated that excess pheomelanin can cause nevus depigmentosus that leads to yellow hair color (Oiso et al., 2018a; Oiso et al., 2018b). Reduced pheomelanin results in the formation of brown hair (Ito et al., 2017; Oiso et al., 2018a; Oiso et al., 2018b). Previous research has reported that black pigment decreases when eumelanin synthesis is blocked, but the range of red and yellow pigment has been shown to be enhanced (Hirobe, Wakamatsu & Ito, 2007; Silvers 1979; Tamate & Takeuchi, 2010). Therefore, we speculated that varying the ratio of pheomelanin to eumelanin plays an important role in the formation of different coat colors in the Rex rabbit.

In the present study, we found that KIT mRNA and protein expression levels were significantly different in the skin of Rex rabbits with different coat colors, the highest observed in black skin, the least in white skin. The results indicate that the ratio of pheomelanin to eumelanin was affected by the expression of KIT, which induced the synthesis of eumelanin. However, KIT mRNA and protein expression levels were not significantly different between a gray and gray-yellow color, indicating that the formation of these two coat colors might not correlate with the synthesis of eumelanin, but instead be determined by pheomelanin. Furthermore, we further confirmed that KIT has a clear effect on melanin-related genes. The results of the present study demonstrate that the mRNA levels of KIT and the melanin-related genes TYR, MITF, PMEL and DCT increased significantly after overexpression of KIT. Conversely, the mRNA levels of these genes clearly decreased after silencing. The results indicate that KIT has a vital role in influencing melanin-related genes at the transcription level.

Melanocyte proliferation and apoptosis are regulated by melanin-related genes and other regulatory interactions. Previous studies have reported that interferon-gamma has a pivotal role in the induction of apoptosis and inhibition of melanogenesis in human melanocytes (Yang et al., 2015). Furthermore, it has been demonstrated that proliferation and melanogenesis in mouse melanocytes can be inhibited by Wnt5a (Zhang et al., 2013). In normal human melanocytes, activation of khellin by ultraviolet A was able to induce melanocyte proliferation and promote melanogenesis (Carlie et al., 2015). In addition, it has been demonstrated that the expression of KIT tyrosine kinase receptor is critical for melanocyte development in human skin (Grichnik et al., 1996). The numbers of melanocytes expressing KIT protein receptor in the skin of patients with vitiligo is lower, indicating that the expression of KIT is related to melanocytes (Norris et al., 1996). Similar results have shown that overexpression of KIT ligand in the epidermis results in melanocytic hyperplasia in transgenic mice (Kunisada et al., 1998). However, whether the function of KIT in regulating the proliferation and apoptosis of Rex rabbit melanocytes is similar to that of humans or mice remains unclear. The results of the present study indicate that melanocyte proliferation was enhanced and melanocyte apoptosis consistently inhibited by overexpression of KIT, indicating that KIT has a positive effect in melanogenesis. In addition, melanin content increased due to melanogenesis. Therefore, we believe that KIT has an important effect on melanocyte proliferation and apoptosis, further regulating melanogenesis.

Conclusions

In conclusion, the KIT gene has an important role in regulating melanocyte proliferation and melanogenesis. KIT mRNA and protein expression levels in the skin of Rex rabbit with black coats were higher than in the skin of other colors, indicating that the KIT gene is involved in the process of creating coat colors, and its expression associated with the production of eumelanin. KIT mRNA expression and that of melanin-related genes was substantially affected by a change in KIT expression (P < 0.01), revealing that the expression of melanin-related genes was regulated. Furthermore, KIT can promote melanocyte proliferation and inhibit apoptosis.

Supplemental Information

Supplemental Information 1 Raw data: KIT cloning, mRNA expression of KIT and melanogenesis-related genes, melanocyte proliferation, apoptosis and melanogenesis

The results indicated KIT positively regulated mRNA expression of melanogenesis-related genes, and promoted melanocyte proliferation and melanogenesis.

Click here for additional data file.

Supplemental Information 2 Raw data: mRNA expression of KIT and melanin-related genes after overexpression and knockdown of KIT, content of melanin, melanocyte proliferation and apoptosis

Click here for additional data file.

Supplemental Information 3 The negative results of WB with antibodies target TYR, MITF, PMEL, and DCT

The raw pictures of “negative result” of WB with antibodies target TYR, MITF, PMEL, and DCT.

Click here for additional data file.

Additional Information and Declarations

Competing Interests

Author Contributions

Animal Ethics

Data Availability

The authors declare there are no competing interests.

Shuaishuai Hu performed the experiments, analyzed the data, authored or reviewed drafts of the paper, and approved the final draft.

Yang Chen conceived and designed the experiments, authored or reviewed drafts of the paper, and approved the final draft.

Bohao Zhao and Naisu Yang performed the experiments, prepared figures and/or tables, and approved the final draft.

Shi Chen performed the experiments, analyzed the data, prepared figures and/or tables, and approved the final draft.

Jinyu Shen conceived and designed the experiments, prepared figures and/or tables, and approved the final draft.

Guolian Bao and Xinsheng Wu conceived and designed the experiments, authored or reviewed drafts of the paper, and approved the final draft.

The following information was supplied relating to ethical approvals (i.e., approving body and any reference numbers):

The experimental procedures was approved by the Animal Care and Use Committee at Yangzhou University (Yangzhou, China, 24 November 2018, No. 201810025). An ordinary housing facility was used and was in keeping with the national standard Laboratory Animal Requirements of Environment and Housing Facilities (GB 14925-2001).

The following information was supplied regarding data availability:

The raw data of KIT cloning, mRNA expression is available in the Supplemental Files and at Figshare: Hu, Shuaishuai (2020): The raw data of KIT cloning, mRNA expression.. figshare. Dataset. https://doi.org/10.6084/m9.figshare.10320104.v2.

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
