# Peer review of "KIT is involved in melanocyte proliferation, apoptosis and melanogenesis in the Rex Rabbit"

_PeerJ, doi:10.7717/peerj.9402_

## Round 0.1 · original submission · Major Revisions

Our reviewers have found many areas of concern: please address them thoroughly in your next revision.

Reviewer 1 ·

Basic reporting

1) English needs to be edited by a native speaker. There are many sentences that are not clearly written or ambiguous.
2) References: There are numerous mistakes in style and content such as author names, year of publication, and so on. In this connection, there are mistakes in citing references in the text, e.g., line 194.
3) Introduction: Please explain why rabbit melanocytes need to be studied regarding the role of KIT when many studied have been done on its roles in mouse and human melanocytes. Any advantages or any specific purposes? Also, what the KIT is needs to be briefly explained along with citation of proper reviews or articles regarding this point, such as a brief summary of signaling pathway in melanogenesis.
4) Discussion is not up to the standard of Peer J. There are many sentences whose meanings are not clear.
5) In Discussion, please discuss how KIT acts on proliferation and melanogenesis in rabbit melanocytes in comparison with mouse and human melanocytes. Are they similar or different?

Experimental design

1) Materials and Methods section is written adequately.
2) Experimental design is acceptable. However, Results section needs to be written more in detail.

Validity of the findings

The authors discussed melanogenesis in term of eumelanin to pheomelanin ratio. However, from the results of this study, discussion in this regard is not fruitful. But speculation is welcome (to be identified as 'speculation').

Additional comments

Two additional minor comments:
1) Some names of Rex rabbits are strange: protein chinchilla and protein yellow. Are they well-known? Any references to color of those Rex rabbits?
2) I believe that scientific names of species (scubas Oryctolagus cuncils) are written in italic.

Reviewer 2 ·

Basic reporting

Hu et al. revealed that KIT could affect the expression of pigmentation-related genes and affect the proliferation and apoptosis of melanocyte. The results are not enough to support their conclusions. The original data are needed. The use of English should be improved.

Experimental design

1. Fig 2. The author only detected the expression of the genes at the transcriptional level. They should also detect the expression at the protein level with WB.
2. Fig3. The author detected the proliferation with CCK-8 and detected the apoptosis with FACS. Only one preliminary screening method does not provide enough proof for the conclusion. The expression of proliferation and apoptosis markers are needed.
3. The authors revealed a phenomenon without any potential mechanism. To make the story complete, they could detect the expression of some potential molecules at the same time, and speculate on the potential mechanism.

Validity of the findings

1. For nearly all the statistical figures, the author only provided the mean and SD. They should provide the original data.
2. Fig 1a. Please provide the uncropped figures

·

Basic reporting

The authors consider the role of KIT in pigmentation of rabbits.They chose to focus on KIT as a pivotal gene of melanin synthesis, modulated its expression in cultured melanocytes and observed the melanogenesis related genes expression after KIT modulation.

The introduction would benefit from more details and references around melanin synthesis and the genes involved. For example, it would be interesting to detail why MITF, TYR, PMEL are studied here, and how their expression is regulated by KIT. The whole melanogenesis being regulated by KIT and MITF, among other molecules, the authors should highlight the connections and regulations of these genes. In this regard, nearly half of the discussion should be moved to the introduction part.

In the discussion (line 220), the authors can not conclude that "KIT directly affected melanin-related genes at the transcription level". The showed a correlation of expression between those genes, but not a direct effect of KIT modulation.

ENglish should be revised, a few typos and grammar issues can be found in the manuscript.

Experimental design

The experimental design is relevant regarding the question defined. However, more experimental details should be provided by the authors.

Concerning the animals, pictures of each coat color should be provided.

Transfection experiments : How did the authors control the transfection efficiency?
how is the expression level of the genes (fig 2 and 3) corrected for transfection efficiency?
How was the transfection performed? in which culture medium?

Quantitative PCR : how did the authors check the quality of RNAs? How much RNA was used for cDNA synthesis? and how much cDNA for RT-PCR? Please provide the primer efficiency and concentration used.

Melanocyte proliferation and apoptosis : were the experimenst performed on several replicates of the transfection?

More generally, for each experiment, the number of samples used and number of replicates (technical, biological) should be precised, as well as the statistics used.

Validity of the findings

The authors should explain the difference and added-value of the present study compared to their precious study (Hu et al., Biochemical Genetics, October 2019, Volume 57, Issue 5, pp 734–744), since this article deals with melanogenesis gene expression in rabbits with different coat colors. Were the seame samples/data used in the two studies?

The authors chose to clone the KIT gene in rabbit. The sequence they described was deposited in Genbank in 2017. SInce then, a detailed and annotated assembly was released (and accessible for example through Ensembl), and should be used by the authors to be compared to their sequence. What does the difference in size account for? Are there variation in nucleotide sequence or protein sequence compared to the reference assembly?

The gene expression of KIT and melanogenesis related genes was investigated in skin. these genes are specific to melanocytes, which are a minor population of skin. Also, in many species, coat color differences are due to gene expression differences and/or differences in amount of melanocytes in the skin. The authors should therefore provide a picture and estimation of quantity of melanocytes in the skins investigated. Without this data, teh authors can not conclude that KIT is less expressed, since the lower level could be due to a lower number of cells where KIT is expressed.
Isn't it possible to quantify the protein signal with the Wes?

Figure 2 : how was annotation of KIT performed?

The authors should comment on the transitory effect of the transfection. They analysed expression 48 or 72 hours after transfection, so how did they ensure i) that the transfection was efficient and ii) that cells were still transfected after 48 or 72 h?
Fig 2c : which siRNA was chosen?

---

## Round 0.2 · Major Revisions

Our reviewers request you to include the replies to the technical questions in the bulk of the paper, rather than only in the SI. They also highlight several areas where their concerns have not yet been addressed. Please go over the manuscript once more and ensure that all requested information is included.

Reviewer 2 ·

Basic reporting

The authors addressed most of the concerns. There are still some concerns remained.

Experimental design

1. The authors claimed that they performed WB with antibodies target TYR, MITF, PMEL, and DCT, but no result was obtained. Why? Can they submit the "negative result" as supplementary or just as raw data?
2. The authors did not respond to one main concern: " The authors revealed a phenomenon without any potential mechanism. To make the story complete, they could detect the expression of some potential molecules at the same time, and speculate on the potential mechanism. "
3. Fig 4a and 4b, the melanin content can be seen directly in the concentrated cells ( Pls refer to DOI: 10.1016/j.jdermsci.2016.04.005), which will make the claim more reliable. Can the author provide the pictures?

Validity of the findings

Some legends are wrong or confused.
1. All the relative expressions were not illustrated clearly. Were they standardized by internal control?
2. The author should quantitatively analyze all of the WB results because the expression of internal control seems apparently different.
3. Fig 3d looks like WB results, but the authors claimed it to be a qPCR results. Fig 3e is apparently not obtained from the KIT-overexpressed group.
4. Fig 4e-4h was wrongly described in the legends.

·

Basic reporting

The authors improved the clarity of the manuscript and structure regarding introduction/discussion.

Experimental design

Many comments were addressed to the authors regarding experimental set-up and results.

Some of the comments were not clearly answered (for example: primer efficiency for the qRT-PCR experiment, efficiency of the transfection, etc...) and must be stated to ensure validity of the findings.

Most comments got, however, a detailed response. Unfortunately, the authors did not include elements of their response in the corrected manuscript. The objective of the reviewers questions is to clarify the article and add essential technical elements to the article. Thus the authors should carrefully add technical details that were asked by the reviewers. For example it is crucial to add the number of animals studied per experiment, in the figure legend, and not only in supplementary data. Also, the differences between the KIT sequence obtained by the authors and the reference sequence should be described and discussed.

The supplementary data are quite complete, although it is difficult to follow; It seems that the numbers attributed to the excel sheets do not correspond to the figures numbers. And there are some missing data (for example, Wes quantification on fig3d)

Finally the authors should review the protocols they described in the Materials and Methods: for example, it is very surprising thet qRT-PCR experiments were performed with 10µM solution of primers. I guess it is the initial concentration of primers. The information that is needed is the final concentration of primers in the PCR reaction.
Similarly, the quantity of siRNAs used or silencing should be given instead of a volume.

Validity of the findings

The results obtained seem relevant and sound. However, if teh authors do not provide the elements asked, the study can not be considered fully validated.

---

## Round 0.3 · Minor Revisions

Please address the final issues, especially regarding the agreement of your figures with the raw data (e.g. confirm that no distortion of the pictures has occurred).

Reviewer 2 ·

Basic reporting

The authors addressed most of the concerns. There are still some concerns remained.

Experimental design

no comment

Validity of the findings

1. It seems that the raw picture of figure 2f-GAPDH and the figure they used in the manuscript is not the same.
2. What do you want to express with Table 4? It seems that there was no data.

·

Basic reporting

Despite typos left, the article is now clear and quite well structured

Experimental design

Most questions regarding experimental design and methods where addressed

Validity of the findings

Fine

Additional comments

The authors addressed the majority of remarks. Their corrections improved the mansucript a lot.

---

## Round 0.4 · accepted · Accept

Thank you for addressing the final points!